# Prognostic Implication of YY1 and CP2c Expression in Patients with Primary Breast Cancer

**DOI:** 10.3390/cancers15133495

**Published:** 2023-07-04

**Authors:** Chihwan David Cha, Seung Han Son, Chul Geun Kim, Hosub Park, Min Sung Chung

**Affiliations:** 1Department of Surgery, Hanyang University College of Medicine, Seoul 04764, Republic of Korea; channyflower@hanyang.ac.kr; 2Department of Life Science and Research Institute for Natural Sciences, College of Natural Sciences, Hanyang University, Seoul 04764, Republic of Korea; todaud0@hanyang.ac.kr; 3Department of Pathology, Hanyang University College of Medicine, Seoul 04764, Republic of Korea; parkhstm@hanyang.ac.kr

**Keywords:** breast cancer, transcription factor, YY1, CP2c, biomarker, prognosis

## Abstract

**Simple Summary:**

YY1 and CP2c is a transcription factor that regulates epigenetic pathways and protein modifications among several kinds of cancer. However, it is still unknown whether YY1 expression has any prognostic significance in patients with breast cancer. Thus, we investigated YY1 expression in association with CP2c in breast cancer patients and their prognostic implications. In this study, quantitative analysis of YY1 and CP2c expression in tumors revealed a negative correlation between them. Patients with YY1-high/CP2c-low expression showed the most favorable survival outcomes. YY1 overexpression was found to be significantly associated with a better prognosis after multivariate analysis. Our study provides novel findings about the association between YY1 and CP2c and its prognostic implication in breast cancer through quantitative analysis at the transcriptome and protein levels.

**Abstract:**

Yin Yang 1 (YY1) is a transcription factor that regulates epigenetic pathways and protein modifications. CP2c is a transcription factor that functions as an oncogene to regulate cell proliferation. YY1 is known to interact with CP2c to suppress CP2c’s transcriptional activity. This study aimed to investigate YY1 and CP2c expression in breast cancer and prognostic implications. In this study, YY1 and CP2c expression was evaluated using immunohistochemical staining, Western blot and RT-PCR assays. Of 491 patients with primary breast cancer, 138 patients showed YY1 overexpression. Luminal subtype and early stage were associated with overexpression (*p* < 0.001). After a median follow-up of 68 months, YY1 overexpression was found to be associated with a better prognosis (disease-free survival rates of 92.0% vs. 79.2%, *p* = 0.014). In Cox proportional hazards model, YY1 overexpression functioned as an independent prognostic factor after adjustment of hormone receptor/HER2 status and tumor size (hazard ratio of 0.50, 95% CI 0.26–0.98, *p* = 0.042). Quantitative analysis of YY1 and CP2c protein expression in tumors revealed a negative correlation between them. In conclusion, YY1 overexpression is a favorable prognostic biomarker in patients with breast cancer, and it has a negative correlation with CP2c at the protein level.

## 1. Introduction

Breast cancer is a highly heterogeneous disease that leads to a variety of diseases prognoses. A combinational histopathologic evaluation for the expression of hormone receptor (HR) and human epidermal growth factor receptor 2 (HER2) affects patient prognosis and treatment decisions [1]. Although several biomarkers have been reported, searching for novel molecular markers that can effectively predict disease progression and prognosis is still of great significance.

Yin Yang 1 (YY1) has regulatory roles in cell proliferation, cell migration, and cell viability [2]. YY1 is known to induce transcriptional activation or repression of many genes associated with cellular differentiation and proliferation [3]. YY1 has been shown to be overexpressed in a variety of cancers including breast cancer, lung cancer, prostate cancer, and ovarian cancer [4,5,6,7]. As a transcription factor (TF), YY1 both activates and suppresses the expression of a number of oncogenes and tumor suppressors involved in various cellular functions, including proliferation, angiogenesis, metastasis, DNA damage response, redox homeostasis, apoptosis, and immunosuppression [8]. In addition to its function as a TF, YY1 is known to act as an adaptor between regulatory RNA and chromatin targets. It binds to nascent mRNA, bridging the mRNA to chromatin [9,10], and promotes enhancer–promoter chromatin loops by forming dimers and promoting DNA interactions [11,12]. Some studies have demonstrated that YY1 overexpression in breast cancer cell lines leads to tumor promotion through the ERBB2 and Akt/Cyclin D1 pathways [13,14]. Another study has shown that YY1 positively induces expression of BRCA1, a tumor suppressor, leading to tumor inhibition in breast cancer [15]. These two facets of YY1, tumor growth promotion versus suppression, remain unclear and further research is needed for the elucidation of prognostic implications.

CP2c (also known as TFCP2, α-CP2, LSF, and LBP-1c) is an evolutionarily conserved TF that is normally expressed ubiquitously at low levels and participates in diverse cellular processes, including cell cycle, immune response, and hematopoiesis [16]. CP2c expression has been shown to be upregulated in cancer cells, including breast cancer, cervical cancer, and hepatocellular carcinoma (HCC) [17,18,19], and levels of CP2c rise in advanced stage among patients with colorectal cancer, and HCC [18,20,21]. CP2c is known to regulate cancer cell proliferation, invasion, angiogenesis, and metastasis [22,23]. Previous research suggested that CP2c and YY1 interact with each other, and their expression is reciprocally regulated in stem cells during spermatogenesis [24,25]. Mechanistically, it was suggested that the HXPR motif of YY1 interacts with CP2c and suppresses the transcriptional activity of CP2c [24]. Several studies reported that the expression and regulatory roles of CP2c and YY1 might affect the development of cancer individually; however, the association between the interaction of these proteins and the prognosis of patients with breast cancer has not yet been investigated.

Thus, this study aimed to investigate the clinicopathological characteristics of YY1 expression and its association with survival outcomes independently and in connection with CP2c expression using formalin-fixed paraffin-embedded (FFPE) tissue blocks from patients with primary breast cancer.

## 2. Materials and Methods

### 2.1. Patients and Clinical Specimens

Consecutive series of clinical information and tissue blocks from 491 patients with primary invasive breast cancer were collected. All patients were diagnosed and underwent curative surgery at Hanyang University Hospital (Seoul, Republic of Korea) between 2002 and 2016. Patients with (1) carcinoma in situ, (2) unknown clinicopathologic information, (3) metastatic disease, (4) previous breast cancer, and (5) no available paraffin blocks were excluded. Clinicopathological data that included the patient’s age, pathologic tumor size, pathologic nodal stage, histological grade, ER/HER2 status, lymphovascular invasiveness, Ki-67 labeling index, and survival outcome were retrospectively collected. All tissue samples were fixed in formalin and embedded in paraffin. All slides stained with hematoxylin and eosin (H&E) together with pathology reports were reviewed. To investigate the correlation between YY1 and CP2c, paired normal tissues from 24 patients were collected. This study was approved by the Institutional Review Board of the Hanyang University Hospital (HYUH 2022-01-029), and the requirement to collect informed consent was waived.

### 2.2. Tissue Microarray Construction

A manual tissue microarrayer (Unitma, Seoul, Republic of Korea) was used for tissue microarray (TMA) construction from archival FFPE tissue blocks. The most representative non-necrotic central portion of the tumor was selected using light microscopy. A tissue cylinder 3 mm in diameter from a previously marked lesion of each donor block was punched and transferred to the recipient block (Unitma, Seoul, Republic of Korea). Each TMA block is comprised of 6 × 5 samples.

### 2.3. Immunohistochemical Staining

Immunohistochemical (IHC) staining for YY1 on 4 μm thick sections from the TMA blocks was performed. All TMA sections were deparaffinized in xylene. The deparaffinized sections were then rehydrated through a series of 5 min washes in 100%, 90%, and 75% ethanol and phosphate-buffered saline (PBS). To retrieve the antigen, the sections were heated in sodium citrate buffer (pH 6.0) in an autoclave at 100 °C for 20 min. Then, endogenous peroxidase activity was blocked using a peroxidase blocking solution (S2023; Dako, Glostrup, Denmark). The TMA slides were incubated with a rabbit monoclonal YY1 antibody (1:100 dilution, ab109237; Abcam, Cambridge, UK), and mouse polyclonal anti-CP2c antibody (610818, BD Biosciences, Franklin Lakes, NJ, USA) at 4 °C overnight and then incubated with a labeled polymer (EnVision/HRP, K5007; Dako, Glostrup, Denmark) for 30 min at room temperature. Monoclonal mouse anti-estrogen receptor (ER), anti-progesterone receptor (PR), anti c-erbB-2, and Ki-67 antibodies (Novocastra Laboratories, Newcastle, UK) were diluted 1:50, 1:100, 1:800, and 1:100 in goat serum, respectively, and 3,3′-diaminobenzidine tetrahydrochloride was used as a chromogen for visualization. The slides were counterstained with Mayer’s hematoxylin.

### 2.4. Interpretation of IHC Staining

YY1 expression was evaluated under a light microscope according to the nuclear staining extent of tumor cells by one pathologist (H.P.) who was blinded to the clinicopathological parameters and the patients’ clinical outcomes. Patients were divided into two subgroups (high or low) according to the mean H-score.

ER status was interpreted using the Allred score for nuclear staining, according to the American Society of Clinical Oncology (ASCO)/College of American Pathologists (CAP) guidelines. Intensity scores of 0, 1, 2, and 3 were given. A score of 0 indicated completely negative intensity, a score of 1 indicated weak intensity, a score of 2 indicated moderate intensity, and a score of 3 indicated strong intensity. A proportion score of 0 to 5 corresponding to percent positive tumor cells was given to 0%, <1%, 1–10%, 11–33%, 34–66%, and 67–100% positive cells, respectively. When the sum of the intensity score and proportion score was between 0 and 2, it was interpreted as negative, and when the sum of these scores was 3 or more, it was interpreted as positive. HER2 status was interpreted according to the ASCO/CAP guidelines. When strong membranous staining was observed in more than 10% of cells via IHC staining, it was interpreted as positive. When weak to moderate membranous staining was observed, dual probe silver in situ hybridization (SISH) was performed. In the SISH, a HER2/CEP17 ratio of 2 or more and a HER2 signal of 4 or more per cell were interpreted as positive. HER2/CEP17 ratio < 2 and average HER2 copy number < 4 were interpreted as negative. In other cases, results were based on 2018 ASCO/CAP recommendations. Ki-67 staining was determined by visually assessing the percentage of cells showing nuclear staining from 0% to 100% in 10% increments.

### 2.5. RT-qPCR from FFPE Tissue

FFPE tissue sections were transferred to 1.5 mL polypropylene microcentrifuge tubes and deparaffinized by incubation at room temperature in xylene for 10 min at 50 °C. After incubation, the tissue was pelleted at 15,000× *g* for 3 min, and the incubation/centrifugation steps were repeated one more time. The deparaffinized tissue pellets were then washed twice with absolute ethanol. The proteins in the FFPE tissues were degraded with 200 µL of protease digestion buffer (20 mM tris-HCl (pH 8.0), 1 mM CaCl_2_, 0.5% SDS) containing 500 µg/mL protease K, followed by incubation for 3 h at 55 °C. Total RNA was isolated using QIAZOL reagent (QIAGEN, Seoul, Republic of Korea, 79306) according to the manufacturer’s procedures. Purified RNA was dissolved in diethyl pyrocarbonate (DEPC) water. Reverse transcription was performed using a High-Capacity cDNA reverse transcription kit (Toyobo, Scottsboro, AL, USA, FSQ-201) in the presence of 400 ng total RNA and 10 pmol random hexamer. RT-qPCR was performed using the SYBR green Master Mix Kit (TaKaRa, San Jose, CA, USA, RR420A). Amplifications were performed using a Light cycler 1.5 real-time PCR system (Roche, Basel, Switzerland). Transcript quantification was calculated as 2(−ΔCt) based on Δ Ct = Δ Ct (treated) − Δ Ct (untreated), with GAPDH transcript levels as internal controls. Errors were calculated from at least two independent experiments. The following oligonucleotide primers were used during RT-PCR: CP2c (Forward, 5′-GGT TGG TGC AGG ACT TTG AT-3′; Reverse, 5′-CAT GGA GTT TCA CTG CTG GA-3′), YY1 (Forward, 5′-GAA TTT GCC AGA ATG AAG CC-3′; Reverse, 5′-TCA TAG CAG AGT TAT CCC TG’3′), and GAPDH (Forward, 5′-TCA GTG GTG GAC CTG ACC TGA CC-3′; Reverse, 5′-TGC TGT AGC CAA ATT CGT TGT CAT ACC-3′).

### 2.6. Western Blot from FFPE Tissue

FFPE tissue sections were transferred to 1.5 mL polypropylene microcentrifuge tubes and deparaffinized by incubation at room temperature in xylene for 10 min. After incubation, the tissue was pelleted at 15,000× *g* for 3 min, and the incubation/centrifugation steps were repeated twice. The deparaffinized tissue pellets were then rehydrated with a graded series of ethanol, briefly air-dried in a fume hood, and weighed. Then, the tissue pellets were homogenized with 100 volumes of protein extraction buffer (20–600 mM Tris–HCl pH 8.0 and 2% SDS). Samples were incubated at 95 °C for 1 h for heat-induced epitope retrieval. The extracts were centrifuged for 15 min at 15,000× *g* at 4 °C, and each supernatant was subjected to SDS-PAGE. For western blotting, electrophoresed proteins were transferred onto polyvinylidene difluoride membranes (GE Healthcare, Chicago, IL, USA, 10600069). The membranes were blocked with 5% BSA in a solution of 0.1% tween 20 and incubated overnight at 4 °C with appropriate dilutions of the following primary antibodies: CP2c (Abcam, ab155238), YY1 (Santa Cruz, Santa Cruz, CA, USA, sc-7341), and ACTB (Santa Cruz, sc-1616). The blots were incubated for 1 h at room temperature with the respective HRP-conjugated secondary antibodies: anti-rabbit IgG-HRP (Ab Frontier, Seoul, Republic of Korea: LF-SA8002) and anti-mouse IgG-HRP (Thermo Fisher, Waltham, MA, USA, 31430). Polyclonal ACTB antibody was used as a loading control for immunoblotting. Proteins were visualized by chemiluminescence using an ECL system (GE Healthcare, RPN2106). Relative amounts of proteins were quantified using Image J software (ver. 1.51).

### 2.7. Statistical Analysis

Pearson’s chi-squared test and Student’s t-test were used to evaluate any potential association between YY1 expression and the clinicopathological parameters in categorical variables. Overall survival (OS) was defined as the duration from surgical treatment to death, and disease-free survival (DFS) was defined as the duration from surgical treatment to the first recurrence or death. The Kaplan–Meier method with a log-rank test was used to construct survival curves and univariate and multivariate Cox proportional hazard ratio models were used to determine the significant prognostic variables. To investigate the correlation between YY1 and CP2c at the mRNA and protein level, Pearson correlation coefficients were calculated. *p* values < 0.05 were regarded as statistically significant. Statistical analysis was performed using R version 3.6.2 (R Foundation for Statistical Computing, Vienna, Austria).

## 3. Results

### 3.1. Patients Characteristics

The clinicopathological characteristics of included patients are summarized in Table 1. The mean age of patients was 52.8 years, and the median follow-up period was 68 months (range, 1–120). Of the included 491 patients, 265 (54.0%) had tumor sizes more than 2 cm and 182 (37.1%) had lymph node metastasis. Of all patients, 174 (35.4%) were in stage I, 220 (44.8%) were in stage II, and 90 (18.3%) were in stage III. About 31% of patients were at a histological grade 3, and 60% of patients were in the HR+/HER2− subtype. Almost 21% of patients were in the TNBC subtype. Among all patients, 63.3% received chemotherapy, and 60.8% had radiotherapy. The mean H-score of YY1 expression was 28.4 and this value was used as a cut-off value to divide patients into two subgroups (high or low YY1 expression). The mean H-score of Cp2c expression was 12.4 and was used as a cut-off as well.

### 3.2. Clinicopathologic Parameters Versus YY1 Expression

YY1 expression was investigated on TMA slides by a single pathologist. All the adjacent normal breast ductal epithelial cells showed intact nuclear YY1 expression. Of the included 491 cases, nuclear YY1 expression in tumor cells was high in 138 cases (28.1%) and low in 353 cases (71.9%) based on IHC staining. Representative microscopic photographs of YY1 and CP2c staining are shown in Figure 1. The association between the nuclear YY1 expression level and clinicopathologic characteristics is shown in Table 2. High YY1 expression was significantly associated with smaller tumor size (*p* = 0.002), lower AJCC stage (*p* < 0.0001), favorable histological grade (*p* < 0.0001), and lower Ki-67 index (*p* = 0.018). High YY1 expression was associated with the HR+/HER2− subtype. There was no significant association between YY1 expression and age, lymph node metastasis, or lymphovascular invasion.

### 3.3. Survival Outcomes Versus YY1 Expression

The impact of nuclear YY1 expression on survival outcomes was investigated. After a median follow-up of 68 months (range 1–120), patients with high YY1 expression showed a better prognosis compared to those with low YY1 expression (OS rates of 96.4% vs. 88.1%, *p* = 0.038 and DFS rates of 92.0% vs. 79.2%, *p* = 0.014, Figure 2). In the meanwhile, there was no difference in survival outcome between subgroups according to CP2c expression (Figure 3).

The univariate Cox regression analysis for DFS revealed primary tumor size (*p* < 0.0001) and high YY1 expression (*p* = 0.016) as the significantly associated prognostic factors. By performing multivariate analysis after adjusting hormone receptor and HER2 status, YY1 overexpression was determined to be an independent favorable prognostic factor for DFS (hazard ratio 0.50, 95% CI 0.26–0.98, *p* = 0.042, Table 3). With regards to OS, YY1 expression was not significant after multivariate analysis.

### 3.4. Correlation Analysis between YY1 and CP2c Expression

It was hypothesized that patients with breast tumors demonstrating YY1 overexpression would show improved prognosis via suppression of the oncogenic function of CP2c. To prove this hypothesis, the correlation between YY1 and CP2c expression in the FFPE samples of patient tumors was investigated. Tumors and paired normal tissues from 24 patients were analyzed. To quantify the expression of YY1 and CP2c at the mRNA and protein level, mRNA and protein were extracted from cancer patient tissue samples and normal tissue samples around cancer tissues and analyzed by RT-qPCR and western blot, respectively (Figure 4).

In the case of patients with YY1 high in the tumor tissue, YY1 expression was also significantly higher in the adjacent normal tissue compared to that of patients with YY1 low at both mRNA and protein levels (Figure 4, A and D). Conversely, the CP2c expression was high only in cancer tissues, and thus the CP2c expression was low in normal tissues at both mRNA and protein levels (Figure 4, B and E). Importantly, the CP2c protein expression in cancer tissues was significantly higher in patients with YY1 low compared to YY1 high (Figure 4, E).

Accordingly, the correlation between CP2c and YY1 expression was analyzed for the data of the tissue type-dependent YY1 expression level using Pearson correlation (Figure 5). The expression of CP2c and YY1 mRNA showed a negative correlation when analyzed not only in cancer tissues with high YY1 expression (*p* = −0.611) but also in all cancer tissues regardless of YY1 expression (*p* = −0.701). In particular, the protein expression of CP2 and YY1 also showed a significant negative correlation in all cancer tissues (*p* = −0.952).

Following these correlations, patients were further divided into four subgroups according to YY1 and CP2c expression levels. As shown in Figure 6, patients with YY1 high and CP2c low expression demonstrated the most favorable prognosis both in terms of OS and DFS.

## 4. Discussion

In the present study, it was found that YY1 overexpression was significantly associated with a better prognosis in patients with primary breast cancer, suggesting that the YY1 expression could be a potential prognostic biomarker. The luminal subtype, early stage, and low tumor grade were significantly associated with overexpression. YY1 overexpression was found to be significantly associated with a better prognosis, and YY1 expression functioned as a favorable prognostic factor after adjustment of HR/HER2 status and tumor size.

These findings are consistent with previous research. Lee et al. (2012) showed that YY1 plays a tumor-suppressive role in breast cancer [15], where YY1 positively regulates the expression of a tumor suppressor BRCA1, leading to tumor inhibition in breast cancer. Another study by Lieberthal et al. (2009) showed that low levels of YY1 induce an invasive breast cancer cell phenotype and that overexpression of YY1 suppresses cancer cell migration by regulating the expression of HP1-alpha [26]. Shen et al. (2019) also reported that YY1-dependent repression of LINC00152 expression leads to elevated PTEN levels and tumor suppression in triple-negative breast cancer [27].

The association analysis between CP2c and nuclear YY1 expression in breast tumor tissues showed that patients with YY1-high and CP2c-low expression showed the most favorable survival outcomes (Figure 4), whereas those with high CP2c expression had a bad prognosis regardless of the nuclear YY1 expression level, suggesting that CP2c is a driver of breast cancer progression. The YY1-based prognostics of breast cancers were affected both in terms of OS and DFS by CP2c expression levels, rendering patients with YY1 high and CP2c low expression to have the most favorable prognosis, suggesting that the combination of YY1 and CP2c expression at protein level could also be used as a prognostic biomarker that can effectively predict the disease progression in patients with primary breast cancer.

These prognostic patterns of YY1 and CP2c expression in breast cancers were also observed in hepatocellular carcinoma (HCC) patients [28], but not in head and neck squamous cell carcinoma (HNSCC) patients [29]. Quantitative analysis of YY1 and CP2c protein expression revealed that HCC patients with high levels of nuclear YY1 and low CP2c expression showed a good prognosis, and high nuclear YY1 expression in HCC patients was more significant than YY1-low expression at a 95% confidence interval [28]. Conversely, protein expression of nuclear YY1 and CP2 showed no association with disease outcome in HNSCC [29]. Importantly, the median OS was decreased for patients with HNSCC demonstrating high YY1 mRNA expression, and patients with a combined high expression of YY1 and CP2 mRNA showed a worse survival. This suggests that unknown underlying mechanisms which regulate mRNA transcription of YY1 and CP2 are the actual cause of worsened survival. It would be suggested that YY1 and CP2c could also counteract at the post-transcriptional levels.

It is known that the expression of YY1 is upregulated in breast cancer tissues as compared to that in adjacent normal tissues [13,30]. However, it is important to note here that YY1 protein expression levels in breast cancers are regulated at both transcriptional and post-transcriptional levels (Figure 3). In the patients with YY1 high tumor tissue, YY1 mRNA levels in breast cancers were significantly correlated with those in the adjacent normal cells, suggesting that the prognostics of breast cancers are primarily determined by the innate YY1 transcription status in the breast tissue. Conversely, YY1 protein levels were not much different between the breast cancers and the adjacent normal tissues, indicating another regulatory mechanism is functional at the post-transcriptional level. Although the CP2c mRNA was upregulated in cancer tissues over that in the adjacent normal tissues, its level was low in YY1 high cancers, which is consistent with previous reports that YY1 directly interacts with CP2c and suppresses the transcriptional activity of CP2c (24). Importantly, the CP2c protein level in YY1 high breast cancers was further downregulated to the CP2c protein level of adjacent normal tissues, which is different from the CP2c-dependent YY1 degradation mechanism through the 20S proteasome pathway (28), suggesting an additional unknown mechanism of YY1-dependent CP2c protein degradation is operational in YY1 high breast cancer tissues. The signal pathway rewiring involving YY1 and CP2c might be important for breast cancer prognostics, and the mechanisms underlying this phenomenon remain to be solved.

There are some limitations to this study. We used a retrospective study design and included cases that were collected from a single institution. As the number of patients with HER2 positive or triple negative subtypes was quite small, the subgroup analysis for these subtypes could not be performed. Further studies are needed to validate our findings and identify candidates in which the combination of YY1 and CP2c expression has a significant impact on treatment decisions such as adjuvant systemic therapy. Despite these limitations, it was confirmed that patients with YY1 overexpression had a favorable prognosis in this study. Our study demonstrates strength in that the quantitative analysis of YY1 and CP2c at the protein level was performed using clinical samples of patient tumors.

## 5. Conclusions

YY1 overexpression is a favorable prognostic biomarker in patients with primary breast cancer, and it has a negative correlation with cofactor, CP2c at the protein level. The combination of YY1 and CP2c expression could be used as a potential biomarker that can effectively predict disease progression. Further research to investigate the signal pathway rewiring YY1 and CP2c expression would be warranted to reveal the underlying mechanisms.

## Figures and Tables

**Figure 1 cancers-15-03495-f001:**
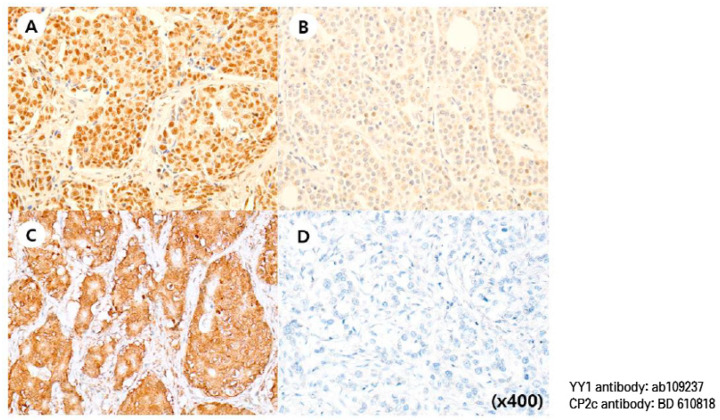
Representative microscopic images of YY1 and CP2c IHC staining. (**A**) high YY1 case (nuclear stain H-score = 180), (**B**) low YY1 case (nuclear stain H-score = 10), (**C**) high CP2c case (cytoplasmic stain H-score = 160), (**D**) low CP2c case (cytoplasmic stain H-score = 0). Rabbit monoclonal YY1 antibody (ab109237; Abcam, Cambridge, UK), and mouse polyclonal anti-CP2c antibody (610818, BD Biosciences) were used.

**Figure 2 cancers-15-03495-f002:**
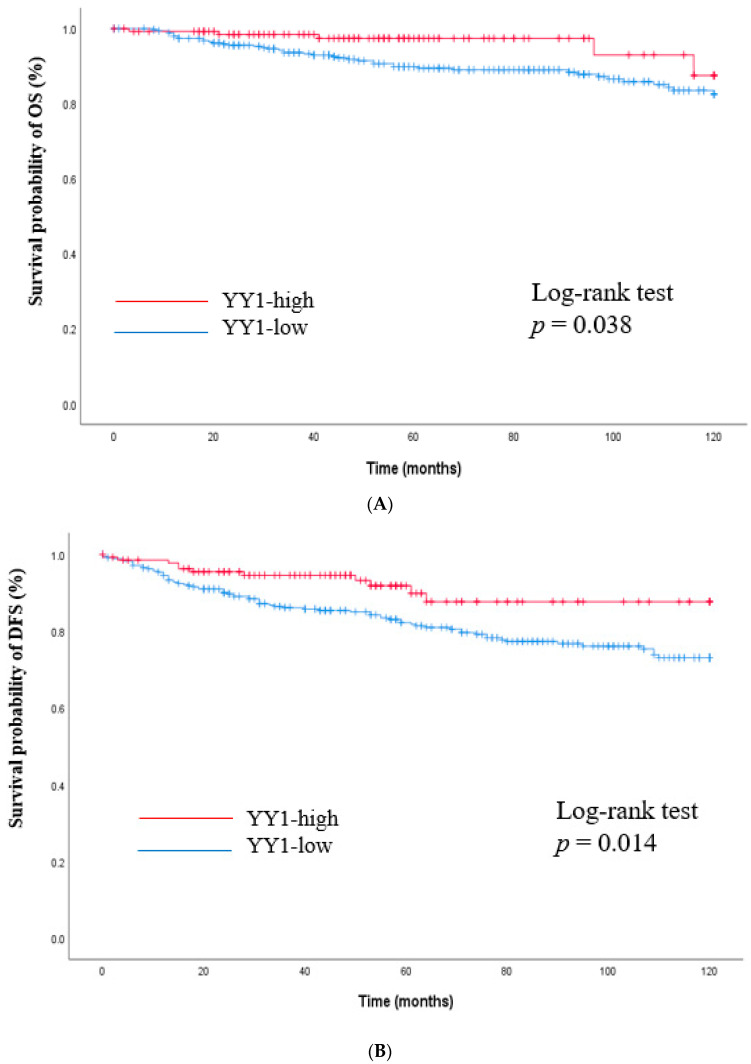
Kaplan–Meier curves for overall survival and disease-free survival according to YY1 expression. (**A**) Overall survival (OS), (**B**) Disease-free survival (DFS). Log-rank test was performed to compare survival outcomes between two groups. *Y*-axis means survival probability (%) of OS and DFS.

**Figure 3 cancers-15-03495-f003:**
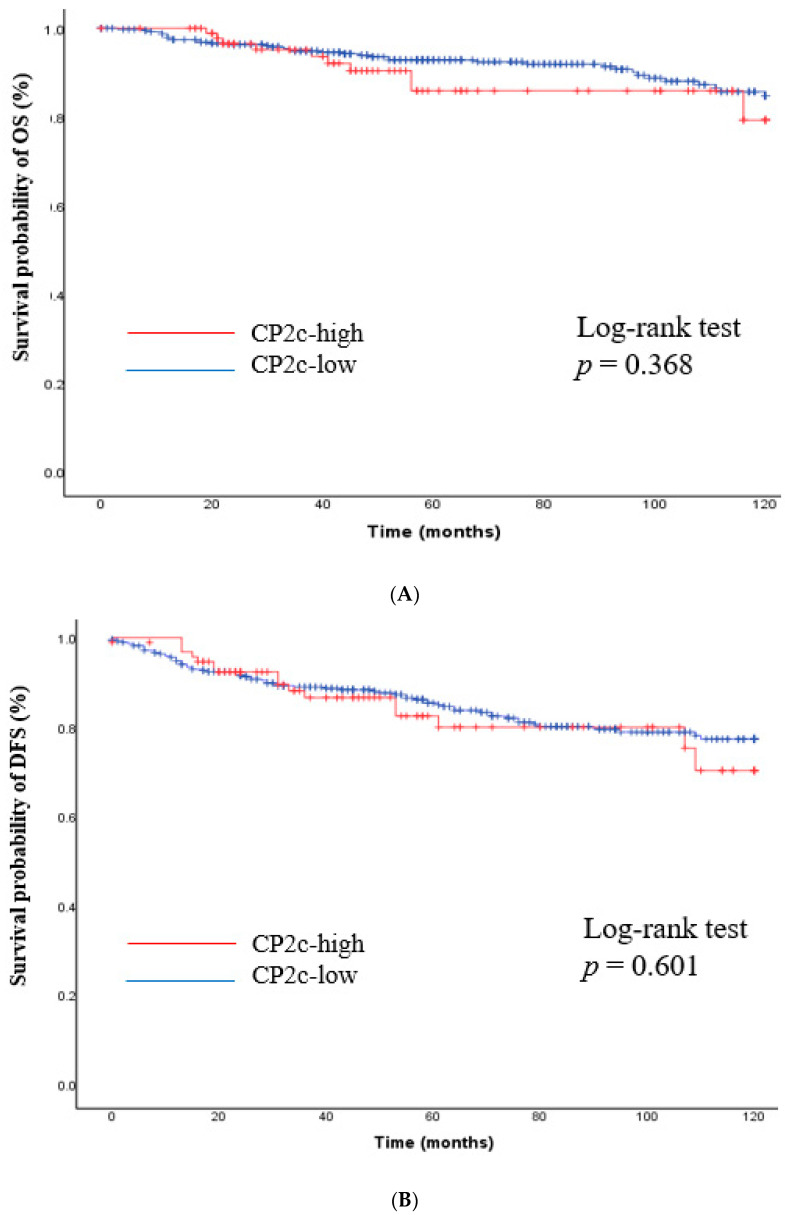
Kaplan–Meier curves for overall survival and disease-free survival according to CP2c expression. (**A**) Overall survival; (**B**) Disease-free survival. Log-rank test was performed to compare survival outcomes between two groups. *Y*-axis means survival probability (%) of OS and DFS.

**Figure 4 cancers-15-03495-f004:**
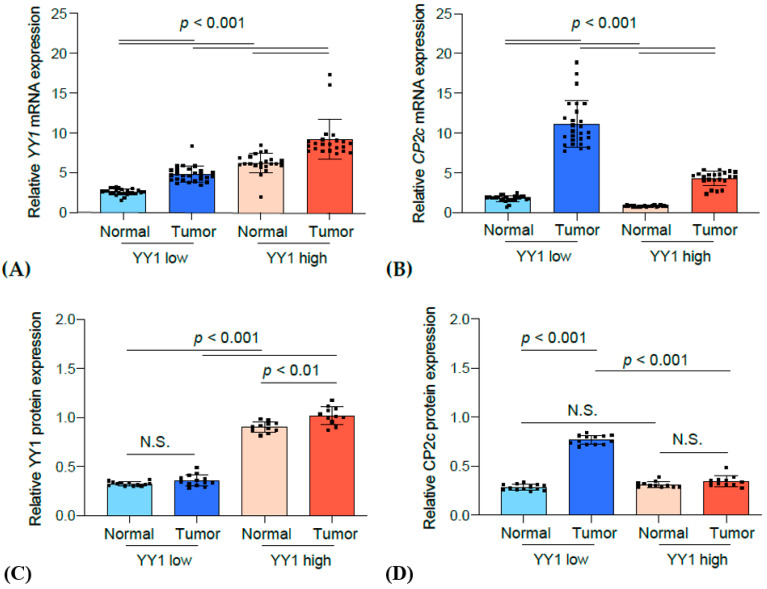
Expression profile of YY1 and CP2c from breast cancer patients’ tissue section. (**A**) YY1 mRNA expression; (**B**) CP2c mRNA expression; (**C**) YY1 protein expression; (**D**) CP2c protein expression. Student’s *t*-tests were employed to assess the statistical significance of differences between data sets.

**Figure 5 cancers-15-03495-f005:**
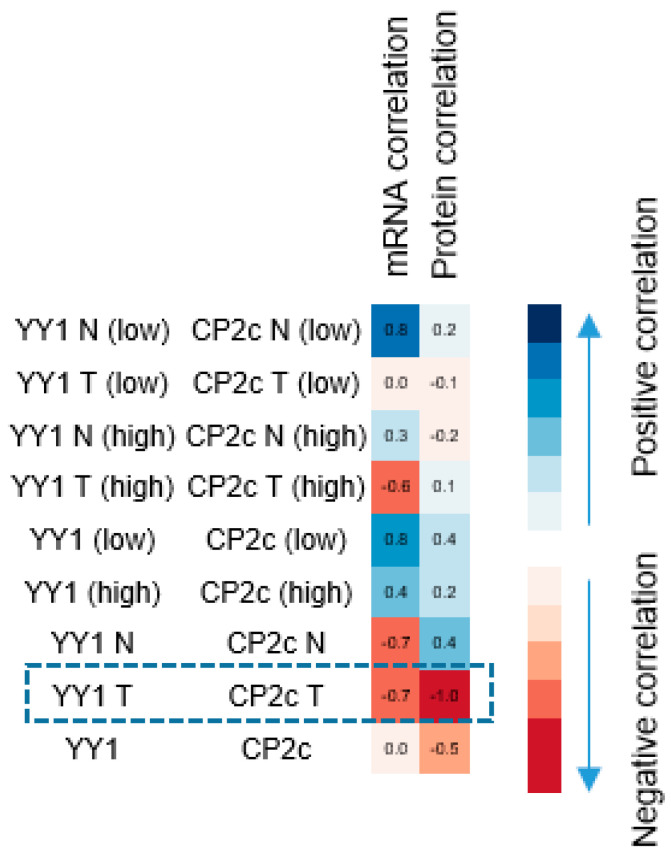
Correlation analysis between CP2c and YY1 expression.

**Figure 6 cancers-15-03495-f006:**
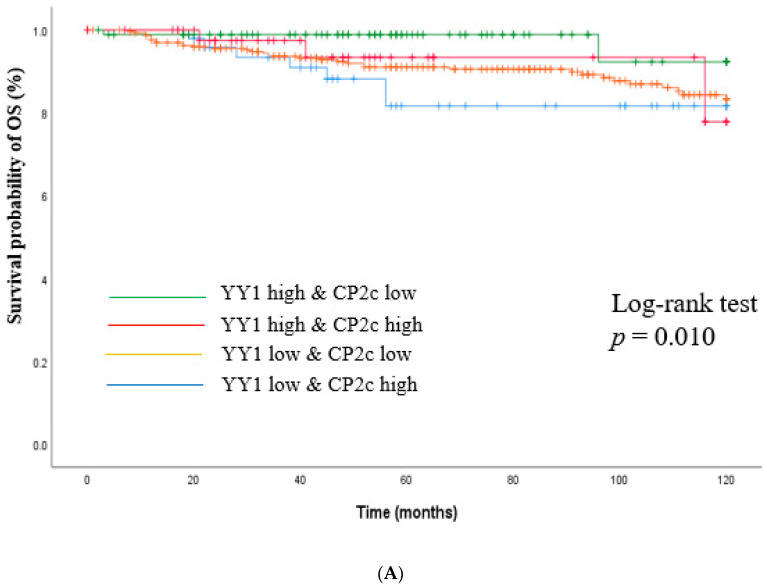
Kaplan–Meier curves for overall survival and disease-free survival according to YY1 and CP2c expression. (**A**) Overall survival; (**B)** Disease-free survival. Log-rank test was performed to compare survival outcomes between YY1 high and CP2c low group vs. YY1 low and CP2c high group. *Y*-axis means survival probability (%) of OS and DFS.

**Table 1 cancers-15-03495-t001:** Baseline characteristics of enrolled patients (*n* = 491).

Variables	Value (%)
Age (years, mean ± SD)	52.8 ± 11.1
Pathologic tumor size	
≤2 cm	220 (44.7)
>2 cm	265 (54.0)
Unknown	6 (1.2)
Pathologic lymph node metastasis	
No	300 (61.1)
Yes	182 (37.1)
Unknown	9 (1.8)
Pathologic nodal stage	
N0	300 (61.1)
N1	109 (22.2)
N2	42 (8.6)
N3	31 (6.3)
Unknown	9 (1.8)
AJCC Stage	
I	174 (35.4)
II	220 (44.8)
III	90 (18.3)
Unknown	7 (1.4)
Histological grade	
1, 2	262 (53.4)
3	151 (30.8)
Unknown	78 (15.9)
Molecular subtype	
HR+/HER2−	294 (59.9)
HER2+	96 (19.6)
TNBC	101 (20.6)
Lymphovascular invasion	
Absence	164 (33.4)
Presence	87 (17.7)
Unknown	240 (48.9)
Ki-67 labelling index	
≤20%	357 (72.7)
>20%	133 (27.1)
Unknown	1 (0.2)
YY1 expression (H-score, mean ± SD)	28.4 ± 49.9

Abbreviations: SD, standard deviation; AJCC, American Joint Committee on Cancer; HR, hormone receptor; HER2, human epidermal growth factor receptor 2; TNBC, triple-negative breast carcinoma; YY1, Yin Yang 1.

**Table 2 cancers-15-03495-t002:** Associations between YY1 expression and clinicopathologic characteristics.

Variables	YY1 Expression		*p* Value
	High (*n* = 138) No. (%)	Low (*n* = 353) No. (%)	
Age (years, mean ± SD)	53.7 ± 10.8	52.8 ± 11.1	0.269
Pathologic tumor size			0.002 *
≤2 cm	77 (56.6)	143 (41.0)	
>2 cm	59 (43.4)	206 (59.0)	
Unknown	2	4	
Pathologic lymph node metastasis			0.051
No	94 (69.1)	206 (59.5)	
Yes	42 (30.9)	140 (40.5)	
Unknown	2	7	
Pathologic nodal stage			0.074
N0	94 (69.1)	206 (59.5)	
N1	23 (16.9)	86 (24.9)	
N2	14 (10.3)	28 (8.1)	
N3	5 (3.7)	26 (7.5)	
Unknown	2	7	
AJCC Stage			<0.001 *
I	67 (49.3)	107 (30.7)	
II	52 (38.2)	168 (48.3)	
III	17 (12.5)	73 (21.0)	
Unknown	2	5	
Histological grade			<0.001 *
1, 2	95 (81.9)	167 (56.2)	
3	21 (18.1)	130 (43.8)	
Unknown	22	56	
Molecular subtype			<0.001 *
HR+/HER2−	116 (84.1)	178 (50.4)	
HER2+	14 (10.1)	82 (23.2)	
TNBC	8 (5.8)	93 (26.3)	
Lymphovascular invasion			0.530
Absence	67 (67.7)	97 (63.8)	
Presence	32 (32.3)	55 (36.2)	
Unknown	39	201	
Ki-67 labeling index			0.018 *
≤20%	111 (80.4)	246 (69.9)	
>20%	27 (19.6)	106 (30.1)	
Unknown		1	

Abbreviations: SD, standard deviation; AJCC, American Joint Committee on Cancer; HR, hormone receptor; HER2, human epidermal growth factor receptor 2; TNBC, triple-negative breast carcinoma; YY1, Yin Yang 1. * *p* < 0.05.

**Table 3 cancers-15-03495-t003:** Univariate and multivariate Cox regression analyses of prognostic factors for disease-free survival.

Variables	Univariate Analysis			Multivariate Analysis		
	Hazard Ratio	95% CI	*p* Value	Hazard Ratio	95% CI	*p* Value
Tumor size						
>2cm (Ref. ≤ 2cm)	3.65	2.12–6.30	<0.0001 *	3.4	1.95–5.91	<0.0001 *
HR status						
Positive (Ref. negative)	0.7	0.45–1.08	0.11		-	
HER2 status						
Positive (Ref. negative)	1.39	0.85–2.29	0.19		-	
YY1 expression						
High (Ref. low)	0.46	0.24–0.87	0.016 *	0.50	0.26–0.98	0.042 *

Abbreviations: CI, confidence interval; Ref, reference; HR, hormone receptor; HER2, human epidermal growth factor receptor 2; YY1, Yin Yang 1. * *p* <0.05.

## Data Availability

The datasets generated during and/or analyzed during the current study are available from the corresponding author upon reasonable request.

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
