# Peer review of "Prognostic Implication of YY1 and CP2c Expression in Patients with Primary Breast Cancer"

_cancers, 2023, doi:10.3390/cancers15133495_

Round 1

Reviewer 1 Report

Minor:

1.     Line 71: it is stated that (and levels of CP2c rise in advanced stage). What are the cancers that show CP2c rise in advanced stage?

2.     Line 95: correct word (sreviewed.)

3.     In Patients and clinical specimens section: have patients had any chemotherapy and/or radiology treatment? If not, please mention that in exclusion criteria?

4.     Line 96: what is meant by this phrase (paired normal tissues from 24 patients were collected). Authors mean that normal tissues are adjacent to tumors OR normal tissues were taken from normal healthy patients. If it is the second scenario, so why the normal samples were taken from normal healthy women? For what purposes?

5.     Line 130-131: what are the score 1 and 2 mean?

6.      In Statistical Analysis section, lines 189-190: it is stated that (disease-free survival (DFS) was defined as the duration from surgical treatment to the first recurrence or death). I think that the disease-free survival (DFS) should be the duration from surgical treatment to the first recurrence of tumour not for patient death. patient death cut point is only used for Overall survival calculation.

7.      In results section, Clinicopathologic parameters versus YY1 expression subsection, lines: 213-214: it is stated that (All of the adjacent normal breast ductal epithelial cells showed intact nuclear YY1 expression). What is the expression in adjacent normal breast tissues? Is it high or low?

8.      Figure 1: it is important  to include images for YY1 and CPc2 expression in normal tissues

9.      In results section, Survival outcomes versus YY1 expression subsection: how many patients were included in analysis? Are the follow-up data available for 491 patients?

10.                         Line 305: what this sentence for? (Log-rank tests for YY1 high & CP2c low group vs. YY1 low & CP2c high group).

Major:

1.     In Immunohistochemical Staining section, lines 107-122: what are the positive and negative controls used for experiment? What measures have been taken to guarantee primary antibody specificity for YY1 and CP2c targets? This is very crucial for immunohistochemistry particularly because investigators used different primary antibodies (different clones and different sources) for immunohistochemistry and western blotting technique to identify YY1 and CP2c protein expressions.

2.     Line 125: why only one pathologist examined the slides? There should be at least two pathologists in order to get reliable and valid scoring results. It is an interpretation process and may be subjected to bias by only one pathologist.

3.     In Interpretation of IHC Staining section, lines: 124-142: it is confusing. First, authors followed H-scoring system forYY1. So what about CP2c. what is the scoring system used for CP2c?. Secondly, why authors used Allred scoring system for ER? Why not H scoring system?. Thirdly, why authors used in situ hybridization (SISH) for staining less than 10%?. What authors mean for (weak to moderate membranous staining was observed) in line 138? Fourthly, what is the scoring system used for other markers like anti-progesterone receptor (PR) and anti c-erbB-2?. Finally, why authors did not standardize and follow one scoring system?

4.     In results section, Survival outcomes versus YY1 expression subsection, lines: 238-243: why investigators did the univariate and multivariate Cox regression analyses of prognostic factors for disease-free survival only? What about overall survival?

5.     In results section, Correlation analysis between YY1 and CP2c expression subsection: how many patients were included in PCR and Western blot analyses? It is very important to mention this information.

6.     In results section, Correlation analysis between YY1 and CP2c expression subsection: investigators should add images showing obtained protein bands for YY1 and CPc2 expression.

Author Response

Response to Reviewer 1 Comments

 Minor:

  1. Line 71: it is stated that (and levels of CP2c rise in advanced stage). What are the cancers that show CP2c rise in advanced stage?

Response: Thank you for your kind comment. As stated in the cited references (# 20, 21), CP2c (also known as LSF) expression was increased among patients with advanced stage of HCC, colorectal cancer. We revised this sentence (line 72).

  1. Line 95: correct word (sreviewed.)

Response: We corrected it.

  1. In Patients and clinical specimens section: have patients had any chemotherapy and/or radiology treatment? If not, please mention that in exclusion criteria?

Response: Among all patients, 63.3% received chemotherapy, and 60.8% had radiotherapy. We added this data in manuscript (line 207-208).

  1. Line 96: what is meant by this phrase (paired normal tissues from 24 patients were collected). Authors mean that normal tissues are adjacent to tumors OR normal tissues were taken from normal healthy patients. If it is the second scenario, so why the normal samples were taken from normal healthy women? For what purposes?

Response: Thank you for comment. We collected paired normal tissues (adjacent to tumors) from 24 patients with breast cancer and examined.

  1. Line 130-131: what are the score 1 and 2 mean?

 Response: Score 1 indicated weak intensity, and score 2 indicated moderate intensity. We added this information (line 132-133).

  1. In Statistical Analysis section, lines 189-190: it is stated that (disease-free survival (DFS) was defined as the duration from surgical treatment to the first recurrence or death). I think that the disease-free survival (DFS) should be the duration from surgical treatment to the first recurrence of tumour not for patient death. patient death cut point is only used for Overall survival calculation.

Response: According to the Clinical Trial Endpoints for the Approval of Cancer Drugs and Biologics from U.S. FDA, disease-free survival is generally defined as the time from randomization until disease recurrence or death from any cause (https://www.fda.gov/media/71195/download). Thank you for your precise comment.

  1. In results section, Clinicopathologic parameters versus YY1 expression subsection, lines: 213-214: it is stated that (All of the adjacent normal breast ductal epithelial cells showed intact nuclear YY1 expression). What is the expression in adjacent normal breast tissues? Is it high or low?

 Response: Adjacent normal breast tissues showed various intensity.

  1. Figure 1: it is important to include images for YY1 and CPc2 expression in normal tissues.

 Response: Representative images for YY1 and CP2c staining in normal tissues were shown below. We added those in supplementary figure 1.

Supplementary figure 1. Representative microscopic images of YY1 and CP2c staining in normal tissues. (A) YY1 expression, (B) CP2c expression.

  1. In results section, Survival outcomes versus YY1 expression subsection: how many patients were included in analysis? Are the follow-up data available for 491 patients?

  Response: All of 491 patients had follow-up data and were included in survival analysis.

  1. Line 305: what this sentence for? (Log-rank tests for YY1 high & CP2c low group vs. YY1 low & CP2c high group).

    Response: We are sorry about it; it should be in the figure legend. It means that log-rank tests were performed between YY1 high & CP2c low group (green line) vs. YY1 low & CP2c high group (blue line) in figure 6.

Major:

  1. In Immunohistochemical Staining section, lines 107-122: what are the positive and negative controls used for experiment? What measures have been taken to guarantee primary antibody specificity for YY1 and CP2c targets? This is very crucial for immunohistochemistry particularly because investigators used different primary antibodies (different clones and different sources) for immunohistochemistry and western blotting technique to identify YY1 and CP2c protein expressions.

Response: Due to the lack of tissue-specific nature of YY1 and CP2c protein, it is challenging to utilize positive and negative control tissues. In this study, we classified the cases into two groups based on the relative differences in expression levels among them. Therefore, even assuming the presence of some non-specific noise, which is expected to have an effect on the entire set of cases, we assumed that this noise would not act as a significant bias negating the observed trends between two groups in our study. Thank you for your precise comments.

  1. Line 125: why only one pathologist examined the slides? There should be at least two pathologists in order to get reliable and valid scoring results. It is an interpretation process and may be subjected to bias by only one pathologist.

Response: Although one pathologist interpreted the staining, the assessment was performed in a blinded manner to the clinicopathological parameters and clinical outcomes of each case, thereby minimizing the potential bias in the results. Additionally, the interpretations of YY1 and CP2c were conducted at time intervals of three months or more, which mitigated the influence of one antibody's interpretation on another, ensuring independent assessment for each antibody.

  1. In Interpretation of IHC Staining section, lines: 124-142: it is confusing. First, authors followed H-scoring system forYY1. So what about CP2c. what is the scoring system used for CP2c?. Secondly, why authors used Allred scoring system for ER? Why not H scoring system?. Thirdly, why authors used in situ hybridization (SISH) for staining less than 10%?. What authors mean for (weak to moderate membranous staining was observed) in line 138? Fourthly, what is the scoring system used for other markers like anti-progesterone receptor (PR) and anti c-erbB-2?. Finally, why authors did not standardize and follow one scoring system?

Response: We appreciate for your kind comment. CP2c was interpreted using the same H-score method as YY1. However, unlike YY1, CP2c was assessed the intensity of cytoplasmic staining. The interpretation of ER, PR, and HER2 followed the ASCO/CAP guideline. The interpretation of ER, PR, and HER2 based on the ASCO/CAP guideline is an internationally standardized procedure in the diagnosis of breast cancer and has been widely adopted in various other studies.

  1. In results section, Survival outcomes versus YY1 expression subsection, lines: 238-243: why investigators did the univariate and multivariate Cox regression analyses of prognostic factors for disease-free survival only? What about overall survival?

 Response: We performed univariate and multivariate regression analysis for overall survival; however, it was not significant for YY1 (as shown below). We added some sentences in line 249-250.

Table. Univariate and multivariate Cox regression analyses for overall survival.

Variables

Univariate analysis

Multivariate analysis

Hazard ratio

95% CI

p value

Hazard ratio

95% CI

p value

Tumor size

> 2cm

(Ref. ≤ 2cm)

3.92

1.83-8.40

<0.0001*

3.3

1.50-7.04

0.003*

HR status

Positive

(Ref. negative)

0.38

0.22-0.68

<0.0001*

0.52

0.28-0.94-

0.031

HER2 status

Positive

(Ref. negative)

1.18

0.60-2.33

0.63

-

YY1 expression

High

(Ref. low)

0.39

0.15-0.98

0.045*

0.6

0.23-1.57

0.296

Abbreviations: CI, confidence interval; Ref, reference; HR, hormone receptor; HER2, human epidermal growth factor receptor 2; YY1, Yin Yang 1. *p <0.05.

  1. In results section, Correlation analysis between YY1 and CP2c expression subsection: how many patients were included in PCR and Western blot analyses? It is very important to mention this information.

Response: Thank you for kind comment. To investigate the correlation between YY1 and CP2c, tumor and paired normal tissues from 24 patients were analyzed. Total of 24 tumor samples (11 and 13 YY1 high and low, respectively, and the normal tissue samples corresponding to each adjacent tissue of patients) were subjected in this analysis (as shown by Western blots in the revised figure 4). We added this information in line 277-278.

  1. In results section, Correlation analysis between YY1 and CP2c expression subsection: investigators should add images showing obtained protein bands for YY1 and CPc2 expression.

 Response: We added images showing protein bands in panel C of revised figure 4 (for Western blot photos). We rearranged panels in figure 4.

Reviewer 2 Report

The manuscript by Cha et al., describes the implication of YY1 and CP2c expression in patients with primary breast cancer. The manuscript is well describing; however, few minor points needs to address before considering the manuscript for further publication.

1. In Figure 1, along with the figure legends please mention the YY1 and CP2c antibody name in the figure also.

2. In Figure 2, 3, and 6, mention the Overall Survival probability and Death free Survival probability in the Y-axis of the figure along with the figure legends.

3. Please mention in each figure legends which statistical analysis was performed to compare the groups.

Author Response

  1. In Figure 1, along with the figure legends please mention the YY1 and CP2c antibody name in the figure also.

 Response: Thanks for comments. We added names of antibodies in revised figure 1.

  1. In Figure 2, 3, and 6, mention the Overall Survival probability and Death free Survival probability in the Y-axis of the figure along with the figure legends.

 Response: As you suggested, we mentioned the survival probability (%) of OS and DFS in the figure along with figure legends.

  1. Please mention in each figure legends which statistical analysis was performed to compare the groups.

 Response: As you pointed out, we mentioned the methods of analysis in each figure legends.

Reviewer 3 Report

General comments

the title "Prognostic implication of YY1 and CP2c expression in patients with primary breast cancer" is very interesting to explore the role of YY1 and Cp2c for better understanding of their specific roles in breast cancer and may contribute to find optional therapeutic agents targeting these molecules.

specific comments

- Page 3 line 130 says "A score of 0 indicated completely negative intensity, and a score of 3 indicated strong intensity" what does it mean score 3 indicated strong intensity? do you mean strong positive intensity?

- Page 3 line 131 says " In the SISH, a HER2/CEP17 ratio of 2 or more and a HER2 139 signal of 4 or more per cell were interpreted as positive" what about the criteria for negative? suggest to be clearly indicated.

- Even though the protein quantification is included in the text, there is not western blot results in figure 4.Highly recommended to be presented the blot with the quantification result.

Author Response

specific comments

- Page 3 line 130 says "A score of 0 indicated completely negative intensity, and a score of 3 indicated strong intensity" what does it mean score 3 indicated strong intensity? do you mean strong positive intensity?

Response: According to ASCO/CAP guideline, intensity score of 0 indicated completely negative intensity, score of 1 indicated weak intensity, score of 2 indicated moderate intensity, and score of 3 indicated strong intensity. As you understood, score of 3 means strong positive intensity for nuclear staining.

- Page 3 line 131 says " In the SISH, a HER2/CEP17 ratio of 2 or more and a HER2 139 signal of 4 or more per cell were interpreted as positive" what about the criteria for negative? suggest to be clearly indicated.

Response: As you suggested, we described clearly about the interpretation of HER2 testing in revised manuscript (line 142-144).

- Even though the protein quantification is included in the text, there is not western blot results in figure 4. Highly recommended to be presented the blot with the quantification result.

Response: Thank you for kind comment. We added images showing protein bands in revised figure 4 (for Western blot photos).

Round 2

Reviewer 1 Report

The current manuscript is fine with me. However, the main concern is that most of patients included in the study had chemotherapy or radiotherapy which may affect the expression of such markers.